# Nicotinamide adenine dinucleotide is transported into mammalian mitochondria

**Antonio Davila[1,2†], Ling Liu[3†], Karthikeyani Chellappa[1], Philip Redpath[4], Eiko Nakamaru-Ogiso[5], Lauren M Paolella[1], Zhigang Zhang[6], Marie E Migaud[4,7], Joshua D Rabinowitz[3], Joseph A Baur[1]\***

[1]Department of Physiology, Institute for Diabetes, Obesity, and Metabolism, Perelman School of Medicine, University of Pennsylvania, Philadelphia, United States; [2]PARC, Perelman School of Medicine, University of Pennsylvania, Philadelphia, United States; [3]Lewis-Sigler Institute for Integrative Genomics, Department of Chemistry, Princeton University, Princeton, United States; [4]School of Pharmacy, Queen's University Belfast, Belfast, United Kingdom; [5]Department of Biochemistry and Biophysics, Perelman School of Medicine, University of Pennsylvania, Philadelphia, United States; [6]College of Veterinary Medicine, Northeast Agricultural University, Harbin, China; [7]Mitchell Cancer Institute, University of South Alabama, Mobile, United States

**\*For correspondence:**
baur@pennmedicine.upenn.edu

[†]These authors contributed equally to this work

**Competing interests:** The authors declare that no competing interests exist.

**Abstract** Mitochondrial NAD levels influence fuel selection, circadian rhythms, and cell survival under stress. It has alternately been argued that NAD in mammalian mitochondria arises from import of cytosolic nicotinamide (NAM), nicotinamide mononucleotide (NMN), or NAD itself. We provide evidence that murine and human mitochondria take up intact NAD. Isolated mitochondria preparations cannot make NAD from NAM, and while NAD is synthesized from NMN, it does not localize to the mitochondrial matrix or effectively support oxidative phosphorylation. Treating cells with nicotinamide riboside that is isotopically labeled on the nicotinamide and ribose moieties results in the appearance of doubly labeled NAD within mitochondria. Analogous experiments with doubly labeled nicotinic acid riboside (labeling cytosolic NAD without labeling NMN) demonstrate that NAD(H) is the imported species. Our results challenge the long-held view that the mitochondrial inner membrane is impermeable to pyridine nucleotides and suggest the existence of an unrecognized mammalian NAD (or NADH) transporter.
DOI: https://doi.org/10.7554/eLife.33246.001

## Introduction

Nicotinamide adenine dinucleotide (NAD) is an essential reduction-oxidation (redox) cofactor as well as a cosubstrate for a growing list of enzymes. Within the mitochondria, NAD accepts electrons from a variety of sources and transfers them to complex I of the electron transport chain, ultimately resulting in the generation of ATP. In addition, NAD serves as a cosubstrate for mitochondrial sirtuins and NAD glycohydrolases (*Dölle et al., 2013*). Mitochondrial NAD levels vary in a circadian fashion and can directly influence fuel selection (*Peek et al., 2013*), as well as determine cell survival under stress (*Yang et al., 2007*). Despite these observations, the mechanisms responsible for generating and maintaining the mitochondrial NAD pool remain incompletely understood.

NAD can be synthesized de novo from tryptophan or via the Preiss-Handler pathway from nicotinic acid, but recycling of the nicotinamide generated by continuous enzymatic cleavage of NAD within the body requires the NAD salvage pathway. This consists of two enzymes: Nicotinamide phosphoribosyltransferase (Nampt), which produces nicotinamide mononucleotide (NMN) in what is

considered the rate-limiting step (*Revollo et al., 2004*), and Nicotinamide mononucleotide adenylyl-transferases (NMNATs), which convert NMN to NAD. Three isoforms of NMNAT have been reported, with NMNAT1 localized to the nucleus, NMNAT2 to the Golgi apparatus and neuronal axons, and NMNAT3 to the mitochondria, providing the first evidence that mitochondria contain some of the machinery to maintain their own NAD pool (*Berger et al., 2005*). Nampt is primarily nuclear and cytosolic, however, a small portion co-purifies with mitochondria from liver (*Yang et al., 2007*). Thus, it was suggested that mitochondria contain a complete NAD salvage pathway and might recycle their own nicotinamide or take it up from the cytosol. Subsequently, Pittelli and colleagues failed to detect Nampt in mitochondria purified from HeLa cells and presented immunofluorescence evidence that it was excluded from the mitochondrial matrix (*Pittelli et al., 2010*). Accordingly, it was proposed that cytosolic NMN is taken up into mitochondria and converted to NAD via NMNAT3 to generate the mitochondrial NAD pool (*Nikiforov et al., 2011*). However, Felici et al reported that the full-length transcript for NMNAT3 is not expressed in HEK293 cells, nor in a variety of mammalian tissues, and that instead the endogenous gene produces two splice variants, one of which produces a cytosolic protein, and the other of which produces a mitochondrial protein involved in NAD cleavage rather than synthesis (FKSG76) (*Felici et al., 2013*). Interestingly, mice lacking NMNAT3 were reported to have defects primarily in erythrocytes, which lack mitochondria, and to have normal NAD levels in heart, muscle, and liver (*Hikosaka et al., 2014*) and normal mitochondrial NAD content in multiple tissues (*Yamamoto et al., 2016*). Felici et al went on to show that providing intact NAD, but not any metabolic precursor, restores the mitochondrial NAD pool in cells where it was depleted by overexpression of FKSG76. They concluded that mitochondria do not synthesize NAD at all, but rather take it up intact from the cytosol, which in turn, can take up NAD from the extracellular space. This interpretation is at odds with recent findings which show that NAD and NMN must first undergo extracellular degradation to nicotinamide, nicotinic acid, or nicotinamide riboside in order to be taken up into cells (*Felici et al., 2013*; *Ratajczak et al., 2016*). Moreover, while yeast and plant mitochondria are known to contain NAD transporters, no mammalian counterparts have been described. Thus, the source of mitochondrial NAD remains to be firmly established.

Here we present evidence that mitochondria directly import NAD. Consistent with previous reports of NMNAT activity in mitochondrial lysates, we find that isolated mitochondria can synthesize NAD from NMN, but not from nicotinamide. However, the majority of this activity is dependent on NMNAT1, which is not mitochondrial, and results in the production of NAD outside of the organelles, rather than filling of the matrix. Using intact myotubes, we demonstrate that isotopically labeled nicotinamide riboside, which is converted to NMN by nicotinamide riboside kinases (NRKs) (*Bieganowski and Brenner, 2004*), contributes directly to the mitochondrial NAD pool without shuttling through an intermediary step as nicotinamide. Substituting labeled nicotinic acid riboside, which generates NAD via cytosolic NAD synthase (bypassing NMN), also results in labeling of mitochondrial NAD, suggesting that fully formed NAD, rather than NMN, is transported.

## Results

### Isolated mitochondria synthesize NAD from NMN, but not from nicotinamide

We initially tested whether NAD levels would increase over time in isolated mitochondria incubated with NAD precursors. In the absence of exogenous metabolizable substrates (state 1, as defined by [*Chance and Williams, 1956*; *Nicholls and Ferguson, 1992*]), warming mitochondria isolated from murine skeletal muscle resulted in a rapid loss of NAD(H) content (data not shown). With the addition of substrate (pyruvate/malate, state 2) and ADP (state 3), the rate of NAD loss was progressively slowed, and co-incubation with NMN, but not nicotinamide or nicotinic acid was found to maintain NAD levels near the starting value (*Figure 1a*). To discern whether increased NAD content in the presence of NMN truly reflected new synthesis, rather than slowed degradation, we held mitochondria in state 2 for 30 min to establish a reduced NAD content, then added ADP (state 3) with or without NMN. Supplementation with NMN restored mitochondrial NAD content in a time and concentration-dependent manner (*Figure 1b–d*). Synthesis of NAD from NMN also appears to be at least partially dependent on membrane potential or ATP production, as addition of the uncoupler FCCP or the complex I inhibitor rotenone significantly attenuated the rate of NAD appearance

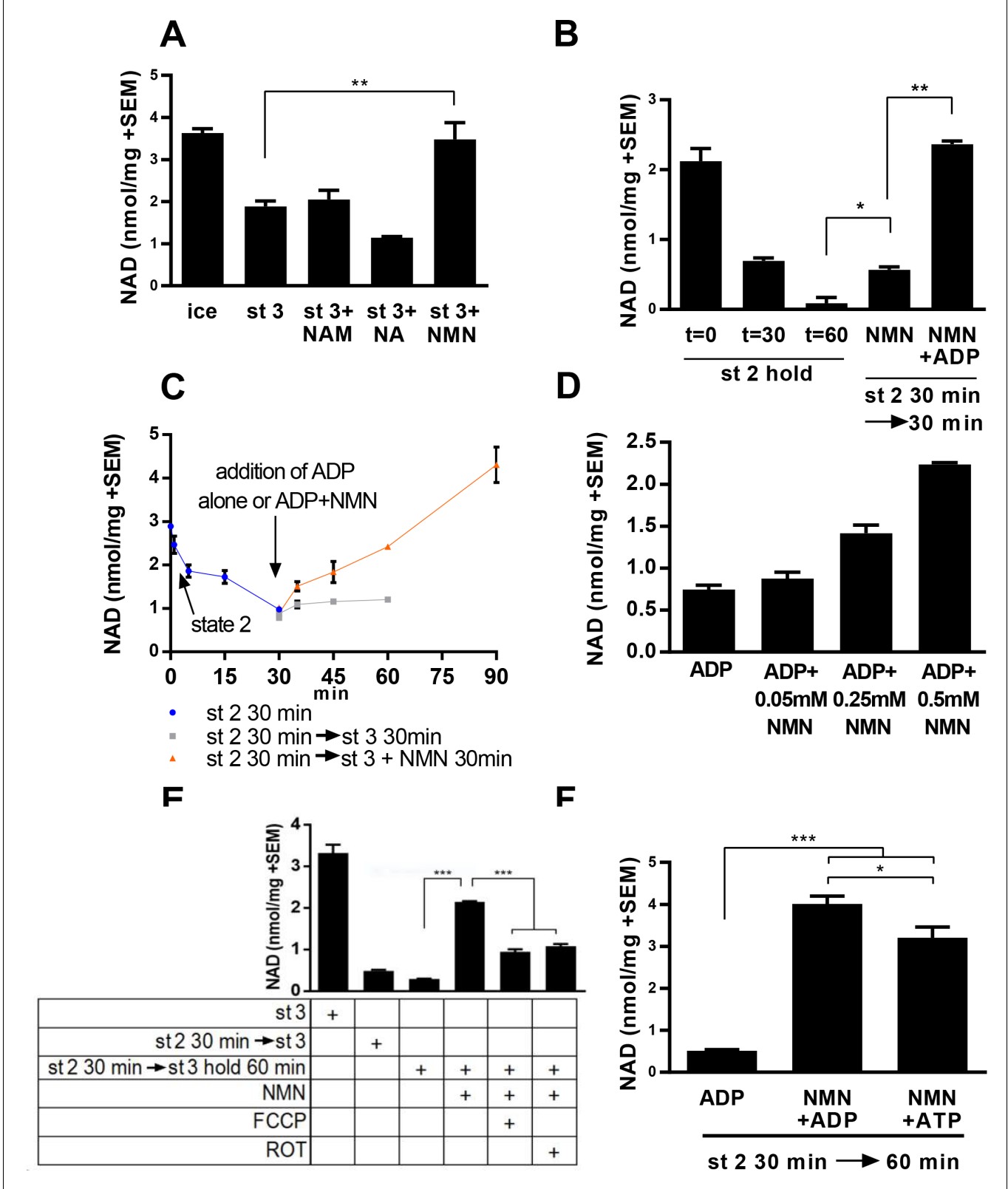

**Figure 1.** Mitochondria synthesize NAD from nicotinamide mononucleotide. (**A**) Mitochondria isolated from murine skeletal muscle were maintained for 30 min at 37°C with shaking in respiratory state 3 (MirO5 respiration buffer containing 10 mM Pyruvate, 5 mM Malate, 12.5 mM ADP) supplemented with 0.5 mM NAM, NA, or NMN. (N = 2–4). (**B**) Mitochondria initially held for 30 min in state 2 (MirO5 respiration buffer containing 10 mM Pyruvate, 5 mM Malate; 37°C with shaking) were then supplemented with NMN alone or NMN + ADP and incubated for an additional 30 min at 37°C. (N = 2). (**C**) Time

*Figure 1 continued on next page*

*Figure 1 continued*

course of mitochondrial NAD levels before and after addition of NMN or NMN + ADP. (N = 2). (D) Isolated mitochondria were held in state 2 for 30 min before adding ADP to stimulate state 3 respiration for 60 min in the presence of increasing amounts of NMN added concomitantly with ADP. (N = 2). (E) Isolated mitochondria were maintained in state 2 at 37°C with shaking for 30 min and then transitioned to state three in the absence or presence of NMN (0.5 mM), FCCP (4 μM), or rotenone (ROT; 0.5 μM) and incubating for an additional 60 min at 37°C with shaking. (N = 4). (F) Isolated mitochondria were held in state 2 for 30 min before being supplemented with NMN alone, NMN + ADP or NMN + ATP and incubated for an additional 30 min at 37°C (N = 2–4). The data shown are means ± SEM from two or more biological replicates, each measured in technical duplicate and are representative of three independent experiments. (*, p<0.05; **, p<0.001; ***, p<0.0001; 2-tailed, unpaired Student's t-test).

DOI: https://doi.org/10.7554/eLife.33246.002

(*Figure 1e*). Consistently, addition of ATP instead of ADP was sufficient to promote NAD synthesis, suggesting that residual ATP present in the mitochondria after isolation was insufficient to support the NMNAT reaction (*Figure 1f*).

In contrast to incubation with NMN, incubation of NAD-deficient mitochondria with nicotinamide did not affect NAD concentration (*Figure 2a*). This was true whether or not exogenous phosphoribo-sylpyrophosphate (PRPP, the second substrate for the Nampt reaction) was supplied. Because the localization of Nampt to mitochondria was described in organelles derived from the liver, we also repeated this experiment with liver-derived mitochondria. Similar to muscle-derived mitochondria, liver-derived organelles synthesized NAD readily from NMN, but were incapable of utilizing nicotin-amide to a measureable degree, whether or not PRPP was provided (*Figure 2b*). To further investi-gate the involvement of NAD salvage enzymes, we employed specific inhibitors of Nampt (FK866) and NMNAT (Gallotannin). As expected, Gallotannin reduced NAD synthesis from NMN (*Figure 2c*), although the incomplete effect was puzzling, given that NMNAT3 is the most Gallotannin-sensitive isoform and is expected to be completely inhibited at this dose (*Berger et al., 2005*). On the other hand, addition of FK866 had no effect, arguing against the possibility that NMN breaks down to nic-otinamide prior to incorporation into NAD.

## Matrix NAD is not restored by NMN treatment in isolated mitochondria

Given that a decline in matrix NAD content will eventually limit respiratory capacity, we next tested whether NMN treatment could restore the respiratory capacity of mitochondria that had been held in state two for an extended period. Despite increasing NAD, NMN treatment did not lead to recov-ery of state three respiration in isolated mitochondria (*Figure 3a*). This suggested two possible inter-pretations: 1) that another form of mitochondrial damage unrelated to NAD content limited respiration, or 2) that the newly synthesized NAD was not localized in the matrix where it would be able to participate in mitochondrial metabolism. To test the latter possibility, we pelleted mitochon-dria after NMN treatment and compared the NAD contents of the pellet and supernatant to the whole mixture. Surprisingly, the increase in NAD was almost exclusively outside of the organelles, with no rescue of matrix NAD content after NMN treatment (*Figure 3b*). We next considered the possibility that mitochondria are sparingly permeable to NAD directly. While low concentrations of NAD failed to have a major impact on matrix NAD content, high (5–10 mM) external NAD led to an appreciable increase. Notably, this concentration is far in excess of whole cell or tissue NAD concen-tration (~300–1000 μM), but is only slightly above our estimates for NAD concentration in the mito-chondrial matrix (3–4 mM, based on the approximation that 1 mg of mitochondrial protein corresponds to ~1 μL of matrix volume [*Das et al., 2003*]). Thus, high external concentrations may be required to create a gradient that favors import. Consistent with these findings, 10 mM external NAD prevented the loss of matrix NAD content over time in mitochondria held in state 2, and signif-icantly slowed the decline in respiratory capacity (*Figure 3c–d*).

## Cytosolic NMN contributes to mitochondrial NAD

To test the behavior of mitochondria in intact cells with physiologically relevant cytosolic concentra-tions of NAD and NMN, we next employed an isotopic labeling approach. Nicotinamide riboside (NR) is taken up by cells and converted to NMN by nicotinamide riboside kinases (NRKs) (*Ratajczak et al., 2016*; *Bieganowski and Brenner, 2004*). We treated intact C2C12 myotubes with NR that had been isotopically labeled on both the nicotinamide ring and the ribose moieties, such

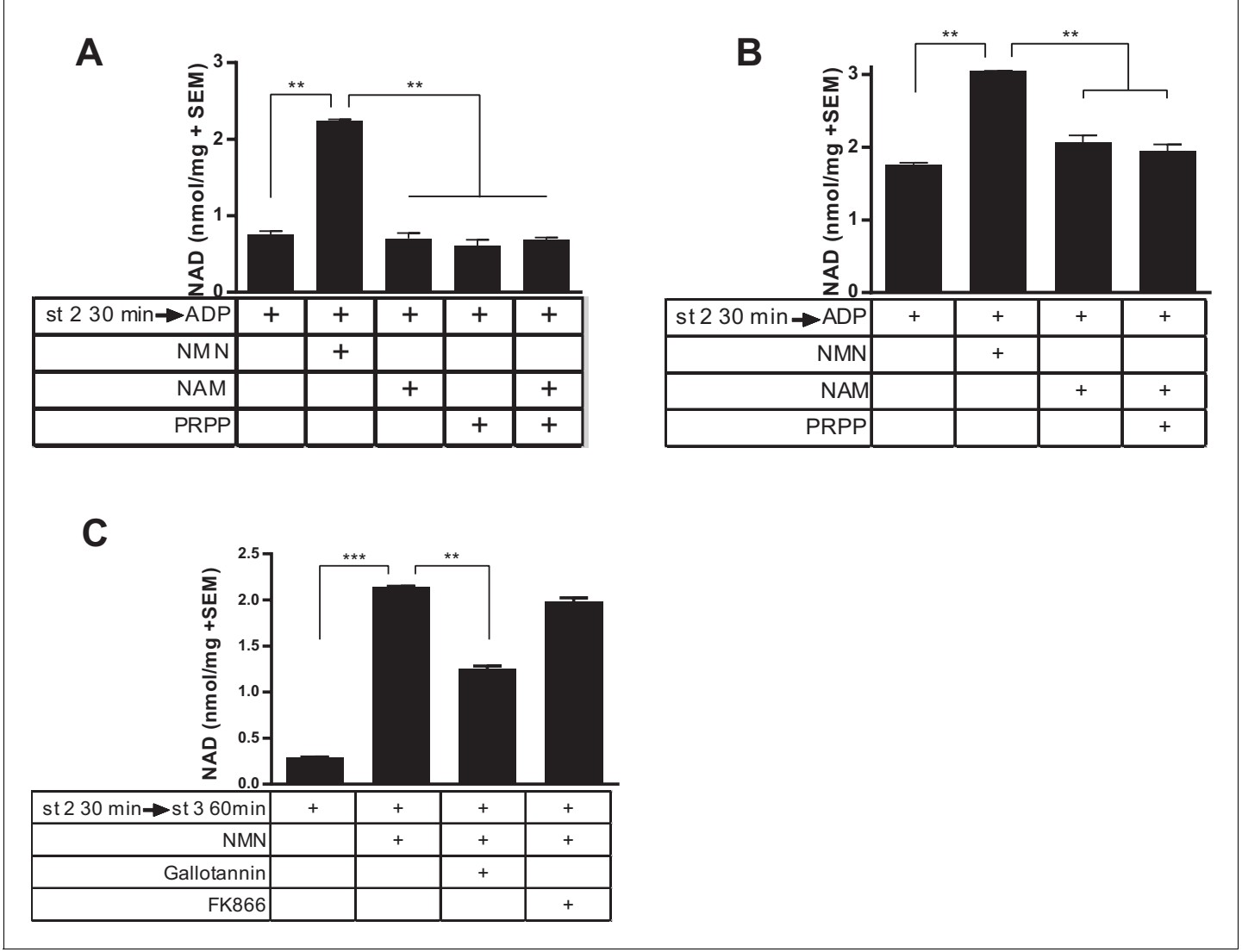

**Figure 2.** Isolated mitochondria do not produce NAD from nicotinamide. (**A**) Mitochondria isolated from murine skeletal muscle were held in respiratory state 2 for 30 min at 37°C with shaking before addition of ADP (state 3) and incubation for 60 min in the absence or presence of the precursors NMN, NAM, PRPP, or NAM and PRPP (0.5 mM). (N = 2). Results are representative of two independent experiments. (**B**) Mitochondria isolated from murine liver were held in state 2 at 37°C for 30 min with shaking before the addition of ADP (state 3) in the presence or absence of 0.5 mM NMN, NAM or NAM and PRPP. (N = 2). (**C**) Muscle mitochondria were maintained in state 2 for 30 min before the addition of ADP (state 3) and further incubated at 37°C for 60 min in the presence or absence of NMN and inhibitors Gallotannin (100 µM) or FK866 (10 nM). (N = 4). (*, p<0.05; **, p<0.005; ***, p<0.0001; 2-tailed, unpaired Student's t-test).
DOI: https://doi.org/10.7554/eLife.33246.003

that its incorporation into NMN and subsequent conversion to NAD would result in retention of both heavy isotopes, whereas degradation of NR by polynucleotide phosphorylase or enzymatic consumption of NAD to generate nicotinamide would separate the labels (*Figure 4a*). We detected a high proportion of doubly labeled NMN and NAD in mitochondria isolated from the myotubes, unequivocally demonstrating that cytosolic NMN contributes to mitochondrial NAD without an intermediated step involving degradation to nicotinamide (*Figure 4b–c*). The slightly more rapid appearance of doubly-labeled NAD in intact whole cell lysates as compared to isolated mitochondria is suggestive that at least some NAD synthesis is occurring outside of the organelles. Since NRK is not present in mitochondria, NMN must be produced in the cytosol, but these data do not allow us to distinguish whether mitochondrial NAD arises from conversion of imported NMN or from direct uptake of cytosolic NAD.

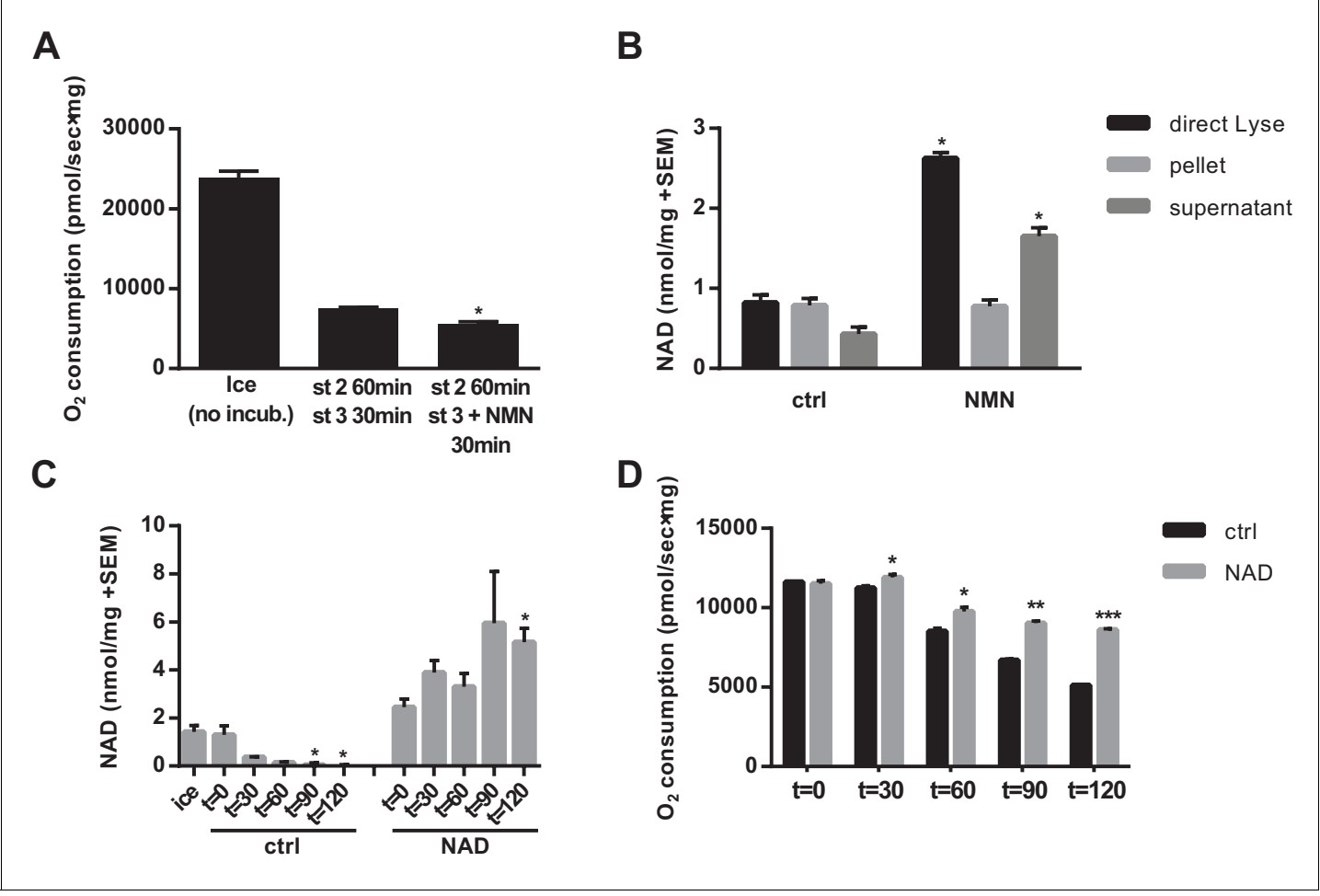

**Figure 3.** NAD synthesized from NMN by isolated mitochondria remains outside of the organelles. (**A**) State three coupled mitochondrial oxygen consumption. Isolated skeletal muscle mitochondria were extracted directly (ice) or maintained in respiratory state 2 (MirO5 respiration buffer containing 10 mM Pyruvate, 5 mM Malate) at 37°C with shaking for 60 min before addition of 12.5 mM ADP (state 3) with or without 0.5 mM NMN and incubated an additional 30 min at 37°C before being measured. (N = 2–3). (**B**) Isolated skeletal muscle mitochondria were maintained in state 2 at 37°C with shaking for 30 min before addition of ADP (state 3) with or without NMN and incubated an additional 60 min. The mitochondrial suspension was then either lysed directly in 0.6M perchloric acid (final concentration) or centrifuged at 10,000 x g for 2 min at 4°C to collect the supernatant and (subsequently washed) pellet which were then extracted with perchloric acid. (N = 2). (**C** and **D**) Isolated skeletal muscle mitochondria were maintained in state 2 at 37°C with shaking with or without 10 mM NAD. At the indicated time points, aliquots were removed from the pooled mitochondrial suspensions and centrifuged to separate the pellet and supernatant (**C**), or analyzed for state three respiratory capacity using high-resolution respirometry (**D**). (N = 2). Results are expressed as mean ±SEM and are representative of three independent experiments. (*, p<0.05; **, p<0.005; ***, p<0.0001; unpaired Student's t-test).

DOI: https://doi.org/10.7554/eLife.33246.004

To discern whether mitochondria have the ability to directly import intact NAD, rather than relying on synthesis from imported NMN, we performed isotopic labeling experiments using nicotinic acid riboside (NAR, *Figure 5—figure supplement 1*). The final step in conversion of NAR to NAD requires the cytosolic enzyme NAD synthase (NADS, *Figure 4a*). Thus, NAD synthesis from this precursor should occur only in the cytosol, and should leave NMN unlabeled. In contrast to this expectation, we found that feeding labeled NAR resulted in nearly equivalent labeling of the total and mitochondrial pools of both NAD and NMN (*Figure 5a–b*). We considered several possibilities to explain the observed NMN labeling: (1) The labeled NAR could have been contaminated with labeled NR, resulting in direct production of both NAMN and NMN, (2) Given the much higher concentration of NAD in cells, non-enzymatic degradation of a small amount of labeled NAD during extraction could account for a substantial portion of the NMN signal, and (3) NMN could be

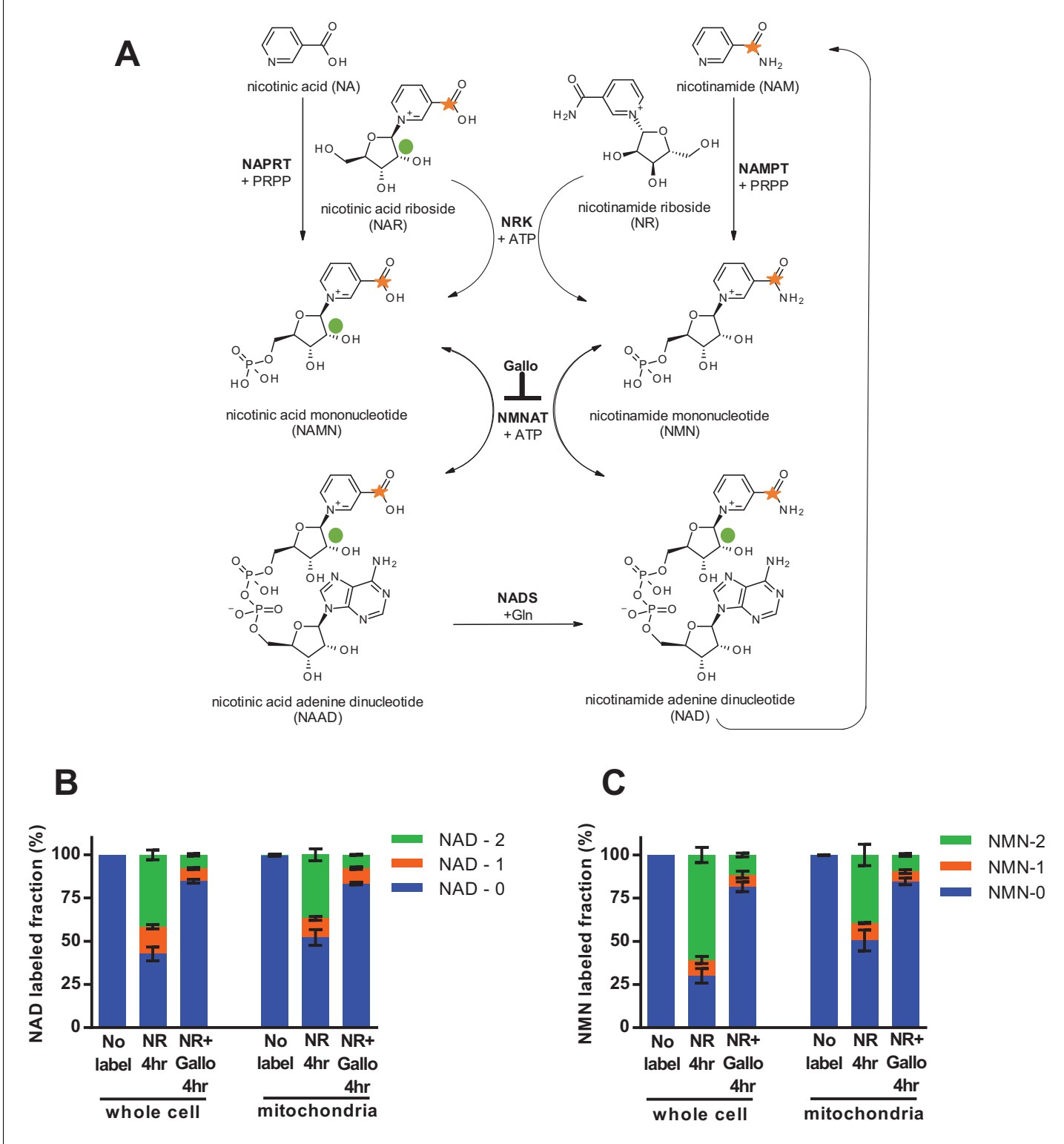

**Figure 4.** Nicotinamide riboside is incorporated intact into mitochondrial NAD. (**A**) Isotopically-labeled nicotinamide riboside (NR) or nicotinic acid (NAR) was synthesized to contain a C-13 on the pyridine carboxyl group and a deuterium on the ribose moiety (NAR labeling shown). (**B**) Fractional labeling of NAD in C2C12 whole cell lysate and isolated mitochondria following 4 hr of incubation with double-labeled NR with or without the NMNAT inhibitor, gallotannin (Gallo; 100 μM). (*No label* group, N = 1; *NR 4* hr group, N = 4; *NR + Gallo 4* hr group, N = 3). (**C**) Fractional labeling of NMN found in C2C12 whole cell lysate and isolated mitochondria following 4 hr of incubation with double-labeled NR with or without gallotannin (Gallo; 100

*Figure 4 continued on next page*

*Figure 4 continued*

μM). (*No label* group, N = 1; *NR 4* hr group, N = 4; *NR + Gallo 4* hr group, N = 3). Data shown are means ± SEM and are representative of 2 independent experiments.

DOI: https://doi.org/10.7554/eLife.33246.005

generated from NAD through enzymatic processes such as reverse flux through NMNATs or degradation of NADH by the mitochondrial Nudix hydrolase Nudt13 and oxidation of the resulting NMNH (*Abdelraheim et al., 2017*; *Long et al., 2017*). To be sure the parent NAR contained no detectable NR contamination we tested it against NR standards; a spike of as little as 0.01 nM NR into 1 μM M NAR was robustly detected, whereas no signal was present in the NAR alone, thereby excluding the first possibility that labeled NMN arose from contaminating NR (*Figure 5c*). Notably, our data also suggest that nicotinic acid-containing nucleotides are not able to enter the mitochondria at all. While NAR was almost undetected by our techniques, we observed a dramatic exclusion of NAMN and NAAD from the mitochondrial fractions (*Figure 5d*). Next we examined the possibility that nonenzymatic degradation of NAD might contribute to the NMN signal. By varying the extraction conditions, we were able to confirm that spontaneous hydrolysis of NAD is a substantial source of contamination of the NMN pool when metabolites are suspended in aqueous solutions (*Figure 5e,f*).

## Cytosolic NAD(H) is imported into the mitochondria

To overcome contamination of the NMN signal by NAD breakdown, we employed two strategies: manipulation of the size of the NMN pool enzymatically, and altering the metabolite extraction protocol to avoid suspension in water. In the first, we used a CRISPR-based system to target each of the three NMNAT isoforms with two independent guide RNAs in C2C12 myoblasts (*Supplementary file 1*). All cell lines differentiated into myotubes with no apparent differences in size or structure at the end of the week-long differentiation protocol. Loss of NMNAT1 protein expression was verified by western blot, while we were unable to reliably detect NMNAT2 or NMNAT3 using available antibodies. However, reduction of mRNA expression and loss of wild type DNA sequence at the target sites were observed in the cell lines (*Figure 6h*; *Figure 6—figure supplement 2*; *Supplementary file 2*) and data not shown). Myotube NAD content was significantly reduced in the two lines targeting NMNAT1, as compared to controls or lines targeting the other isoforms (*Figure 6a*). There were no significant differences in mitochondrial NAD content in freshly isolated organelles (*Figure 6b*, ice). However, organelles from the NMNAT1-targeted lines showed increased susceptibility to NAD depletion by holding in state 2, and limited ability to synthesize NAD from NMN (*Figure 6b*), supporting the model that the majority of NMNAT activity in mitochondrial preparations arises from contaminating NMNAT1, rather than matrix-localized NMNAT3. This conclusion was further supported by proteinase K treatment of mitochondrial preparations. Limited digestion removes proteins in the supernatant or on the outer mitochondrial membrane, while leaving matrix proteins intact. Accordingly, proteinase K reduced conversion of NMN to NAD without affecting respiratory capacity (*Figure 6—figure supplement 2*). In addition to reduced NAD concentration, myotubes with NMNAT1 targeted had dramatically increased NMN content in whole-cell lysates, consistent with a major role for this isoform in NAD synthesis from NMN (*Figure 6c*). Rescuing NMNAT1-targeted cells with human NMNAT1, which does not contain the targeted sequence, attenuated the increase in NMN concentration, and enhanced NAD synthesis from NMN by isolated mitochondria (*Figure 6—figure supplement 3*). In contrast, targeting NMNAT2 or NMNAT3 did not lead to obvious changes in pyridine nucleotide distribution (*Figure 6d*). As hoped, treatment of NMNAT1 targeted cells with NAR led to a large discrepancy in the fractional labeling detected for the total NMN and NAD pools (*Figure 6e*). This is consistent with dilution of any NMN signal that resulted from NAD hydrolysis into the larger pool of unlabeled NMN, but also consistent with the possibility that flux through NMNAT1 contributed to some labeling of NMN in wild type cells. In either case, the higher labeling of NAD as compared to NMN allowed us to resolve the source of mitochondrial nucleotides. The fractional labeling of mitochondrial pyridine nucleotide pools clearly approached that of NAD and far exceeded that of (whole cell) NMN (*Figure 6f,g*). This labeling pattern can only be explained if NAD is taken up directly by mitochondria in NMNAT1-targeted cells. Thus, our findings

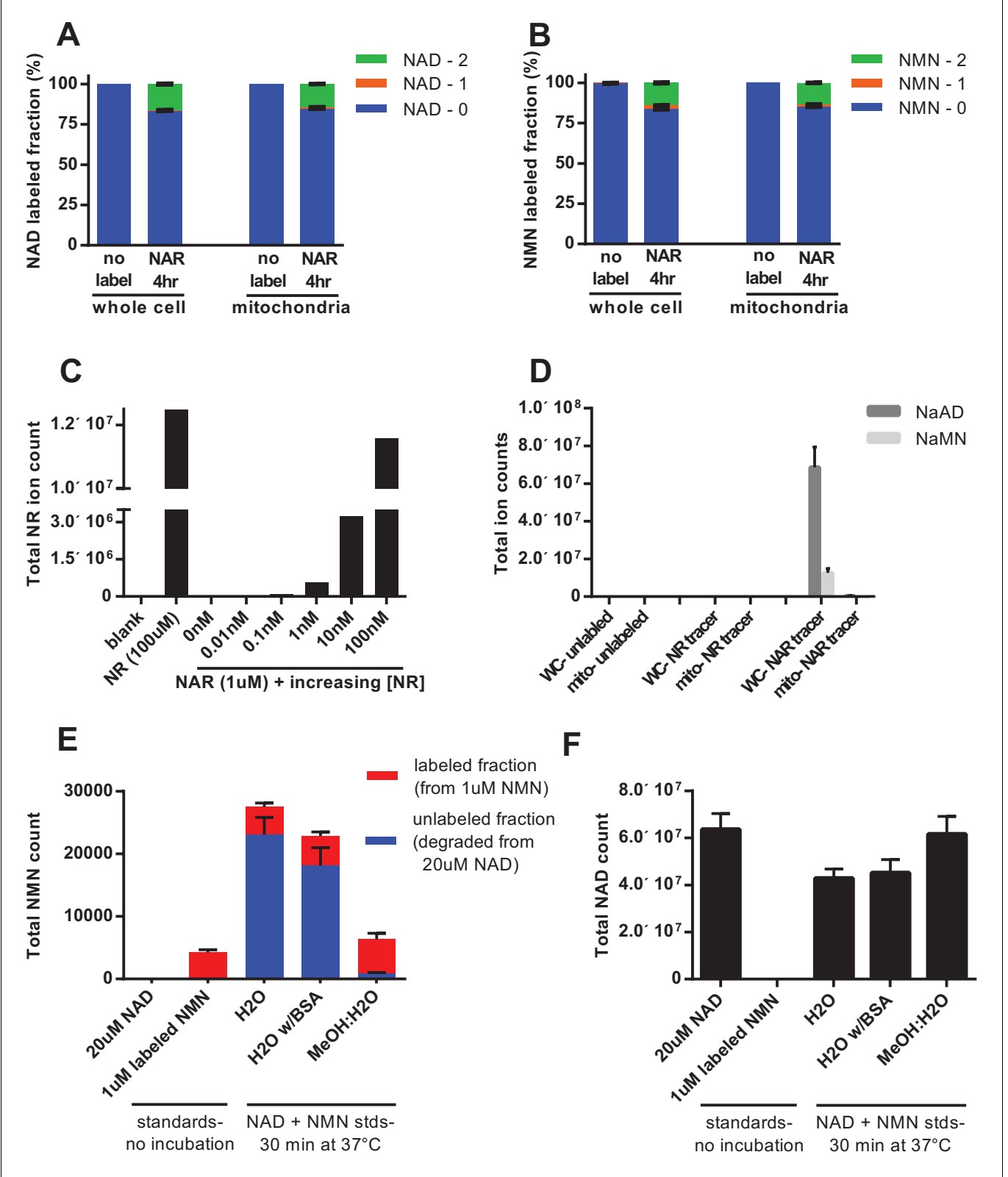

**Figure 5.** Nicotinic acid riboside is incorporated intact into mitochondrial NAD. (**A**) Fractional labeling of NAD in C2C12 whole cell lysates and isolated mitochondria following 4 hr of incubation with doubly-labeled NAR. (N = 5). (**B**) Fractional labeling of NMN in C2C12 whole cell lysates and isolated mitochondria following 4 hr of incubation with doubly-labeled NAR. (N = 5). (**C**) Confirmation of the lack of NR contamination in NAR. 1 µM NAR was combined with increasing concentrations of NR (0–100 nM) to demonstrate that NR is absent in the NAR and readily detected by this methodology.
*Figure 5 continued on next page*

*Figure 5 continued*

(Single measurements). (**D**) Total ion counts for NAAD and NAMN in whole cell lysates and mitochondrial isolates from differentiated C2C12 cells treated with isotopically-labeled NR or NAR tracers for 4 hr. Results expressed as means ± SEM. (N = 3). (**E**) Incubation of NAD at 37°C in water, but not 80% methanol results in substantial degradation to NMN. Blue bars show unlabeled NMN resulting from degradation from a 20 μM NAD standard spike; Red bars indicate labeled NMN from spiked-in standard (1 μM, dual labeled). (N = 3). (**F**) NAD total ion count measured in parallel from same samples in (**E**). (N = 3).

DOI: https://doi.org/10.7554/eLife.33246.006

The following figure supplement is available for figure 5:

**Figure supplement 1.** Characterization of doubly-labeled nicotinic acid riboside.

DOI: https://doi.org/10.7554/eLife.33246.007

indicate that mitochondrial pyridine nucleotides originate from imported NAD (or NADH), rather than import of cytosolic NMN.

In a parallel approach, we were able to demonstrate that by injecting methanolic extracts directly into the LC-MS without a drying/concentration step, we were able to completely avoid the artifactual hydrolysis of NAD to NMN (*Figure 7a,b*). Repeating the NAR treatment using this method revealed very low NMN levels in the whole cell lysates with almost no detectable labeling, whereas the fractional labeling of NAD was consistent with that in previous experiments (*Figure 7c*). Mitochondria isolated from these cells contained labeled NAD, confirming that they import fully synthesized NAD from the cytosol. NMN in mitochondria was also labeled, and we speculate that this reflects degradation of a small proportion NAD during the isolation process. Similarly, treating the human cell lines HEK293 and HL-60 with doubly-labeled NAR resulted in the appearance of doubly-labeled NAD within the mitochondria in the absence of detectable NMN (*Figure 7d*). Taken together, our experiments confirm that despite the lack of any recognized transporter, mammalian mitochondria, like their yeast and plant counterparts, are capable of importing NAD(H).

## Discussion

Mammalian mitochondria lack obvious homologues of the NAD transporters found in yeast and plant mitochondria, raising the question of how they are able to obtain the cofactor. Evidence has been presented in support of direct NAD uptake (*Felici et al., 2013*; *Rustin et al., 1996*), or intramitochondrial synthesis from nicotinamide (*Yang et al., 2007*; *Kun et al., 1975*) or NMN (*Nikiforov et al., 2011*). Our current results support the model that direct uptake of intact NAD contributes to the mitochondrial NAD pool. However, we note that we cannot exclude further contributions from intramitochondrial NAD synthesis.

Yang et al. showed that a portion of Nampt co-purifies with mitochondria from liver, suggesting the model that mitochondria contain an intact NAD salvage pathway, and take up nicotinamide, rather than NMN or NAD (*Yang et al., 2007*). This proposal is consistent with earlier work by Grunicke and coworkers showing that [14]C-labeled nicotinamide incubated with isolated mitochondria is incorporated into both NMN and NAD (*Kun et al., 1975*). However, it is possible that an exchange reaction catalyzed by NAD glycohydrolases (or sirtuins), rather than net biosynthesis could have been responsible for the labeling observed in these experiments (*Behr et al., 1981*). Moreover, we were not able to observe net NAD synthesis when isolated mitochondria were supplied with nicotinamide, with or without exogenous PRPP. Importantly, we cannot exclude that PRPP might need to be generated within the mitochondrial matrix, or that mitochondrial Nampt activity might be present in certain cell types or under certain stresses. However, the present data do not support the ability of mitochondria to synthesize NAD autonomously from nicotinamide, and we note that neither Nampt nor PRPP synthetase has been reported as a mitochondrial protein in the recently updated MitoCarta2.0 database (*Calvo et al., 2016*).

The mitochondrial localization of NMNAT3 strongly suggests that the organelles might be capable of taking up and using NMN from the cytosol when required. In agreement with previous studies detecting NMNAT activity in mitochondrial lysates (*Berger et al., 2005*; *Barile et al., 1996*; *VanLinden et al., 2015*; *Yahata et al., 2009*), we demonstrate that isolated mitochondria synthesize NAD from NMN. However, the use of intact organelles in our experiments allowed us to discern that the vast majority of, if not all NAD generated from NMN by isolated mitochondria ends up outside

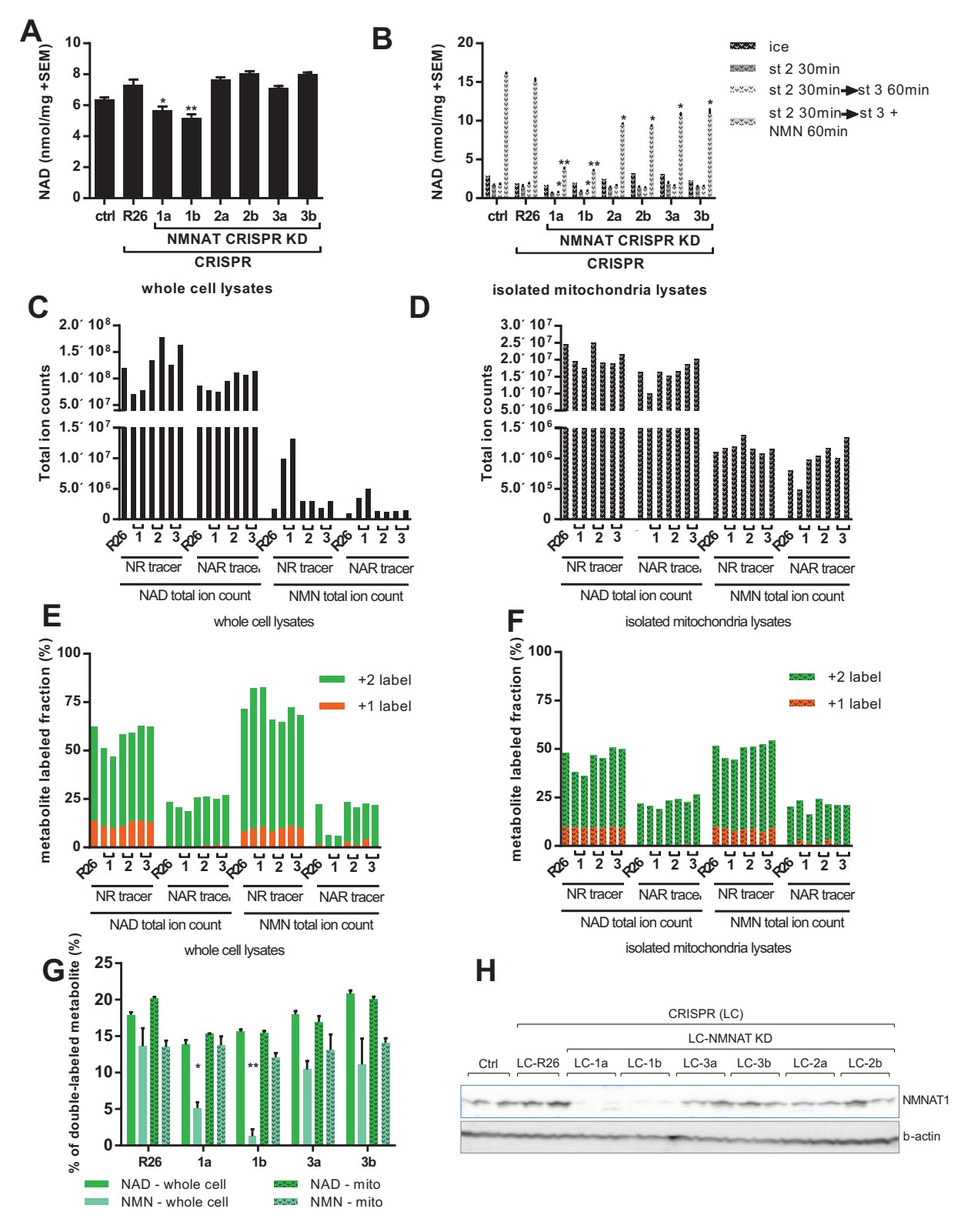

**Figure 6.** Labeling of mitochondrial NAD tracks that of total NAD, but not of total NMN. For all panels, data representing whole cells are depicted as solid bars, whereas data from isolated mitochondria are shown with a stippled pattern. (**A**) Differentiated C2C12 parental and LentiCRISPR transgenic myotubes were analyzed for NAD content. The cells are as follows: ctrl- parental line with no vector; R26- vector control; 1a and 1b- two separate guide RNAs targeting NMNAT1; 2a and 2b- two separate guide RNAs targeting NMNAT2; 3a and 3b- two separate guide RNAs targeting NMNAT3. (N = 3).
*Figure 6 continued on next page*

*Figure 6 continued*

(**B**) Mitochondria isolated from differentiated C2C12 cells were held in state 2 (MirO5 respiration buffer containing 10 mM Pyruvate, 5 mM Malate) at 37°C with shaking for 30 min. They were then collected and lysed in perchloric acid immediately, or transitioned into state three by adding ADP (12.5 mM, final concentration) with or without supplementation with NMN (0.5 mM, final concentration) and maintained for 60 min at 37°C with shaking before collection. (N = 2–4). (**C–D**) Total ion counts for NAD and NMN in extracts from C2C12 LentiCRISPR whole cells (**C**) and isolated mitochondria (**D**) following a 4 hr incubation with isotopically-labeled NR or NAR tracer. (Single measuements). (**E–F**) Fractional labeling of metabolites (NAD and NMN) measured in C2C12 LentiCRISPR whole cells (**E**) and isolated mitochondria (**F**) after a 4 hr incubation with isotopically-labeled NR or NAR tracer. (Single measurements). (**G**) Fractions of double-labeled NAD and NMN measured in C2C12 LentiCRISPR whole cell and mitochondrial lysates following 4 hr incubation with isotope-labeled NAR (means ± SEM). (N = 3). (**H**) Immunoblot confirming NMNAT1 knockout in CRISPR C2C12 cell line. (*, $p<0.05$; **, $p<0.001$; 2-tailed, unpaired Student's t-test versus R26).

DOI: https://doi.org/10.7554/eLife.33246.008

The following figure supplements are available for figure 6:

**Figure supplement 1.** mRNA expression of NMNAT isoforms after CRISPR targeting.
DOI: https://doi.org/10.7554/eLife.33246.009

**Figure supplement 2.** Partial digestion of mitochondrial preps with proteinase K impairs NAD synthesis, but not respiration.
DOI: https://doi.org/10.7554/eLife.33246.010

**Figure supplement 3.** Overexpressing human NMNAT1 decreases NMN accumulation in NMNAT1 targeted cells and enhances the ability of mitochondrial preps to synthesize NAD from NMN.
DOI: https://doi.org/10.7554/eLife.33246.011

the matrix. Moreover, the bulk of this activity is lost when mitochondria are isolated from myotubes lacking the nuclear isoform NMNAT1, suggesting that it arises from small amounts of contamination in the mitochondrial preparations. Since NMNAT1 has been localized exclusively to the nucleus (*Berger et al., 2005*; *Zhang et al., 2012*), our observations most likely reflect nuclear contamination rather than association of NMNAT1 with the mitochondrial outer membrane. It is tempting to speculate that this might also account for the observation of Felici et al. that their mitochondrial lysates contained NMNAT activity that was not attributable to any transcript of the *Nmnat3* gene (*Felici et al., 2013*). Therefore, our data on isolated mitochondria do not provide direct evidence for the ability of NMN import to contribute to mitochondrial NAD.

While mammalian mitochondria are generally considered to be impermeable to pyridine nucleotides (*Stein and Imai, 2012*; *Chappell, 1968*), at least two studies have previously reported evidence for uptake of NAD. Rustin et al. reported that direct addition of NAD restored mitochondrial NAD levels and respiration rate in digitonin-permeabilized human cells that had reduced NAD content due to extended culture without medium changes (*Rustin et al., 1996*), although it is not clear that a rapid breakdown and resynthesis could be completely excluded in these experiments. Felici et al. reported evidence that in HEK293 cells, brain, skeletal muscle, and kidney, the full length transcript described for NMNAT3 does not exist, and that instead, two splice variants are detectable, encoding a cytosolic protein and the mitochondrial protein FKSG76, neither of which is translated at detectable levels (*Felici et al., 2013*). This observation is underscored by the lack of obvious phenotypes in most tissues of mice lacking the NMNAT3 gene, with the exception of erythrocytes, which have cytosolic NMNAT3 and no mitochondria (*Hikosaka et al., 2014*). Interestingly, overexpression of FKSG76 depletes NAD from the mitochondria of intact cells. These depleted mitochondrial NAD levels can be rescued by exogenous NAD, but not by any precursor, leading the authors to propose that intact NAD crosses the plasma membrane and subsequently enters the mitochondria directly. Notably, the NAD precursors provided by Felici et al. should all be incorporated into the nucleocytosolic pool of NAD, and thus would be available to replenish mitochondrial NAD via direct transport. The lack of rescue of FKSG76-depleted mitochondrial NAD levels after precursor treatment therefore implies that either import of precursors or NAD synthesis from them is too slow to compete with the degradation mechanism, whereas direct NAD influx is rapid, or that other aspects of NAD synthesis are impaired in these cells. In addition, it has been reported by Nikiforov et al. that pyridine nucleotides are not transported across cell membranes efficiently and are instead broken down to the corresponding nucleosides or further before being taken up (*Nikiforov et al., 2011*). This model is distinctly at odds with the finding of Felici et al. that extracellular NAD but not nicotinamide riboside is able to restore mitochondrial NAD in cells overexpressing FKSG76.

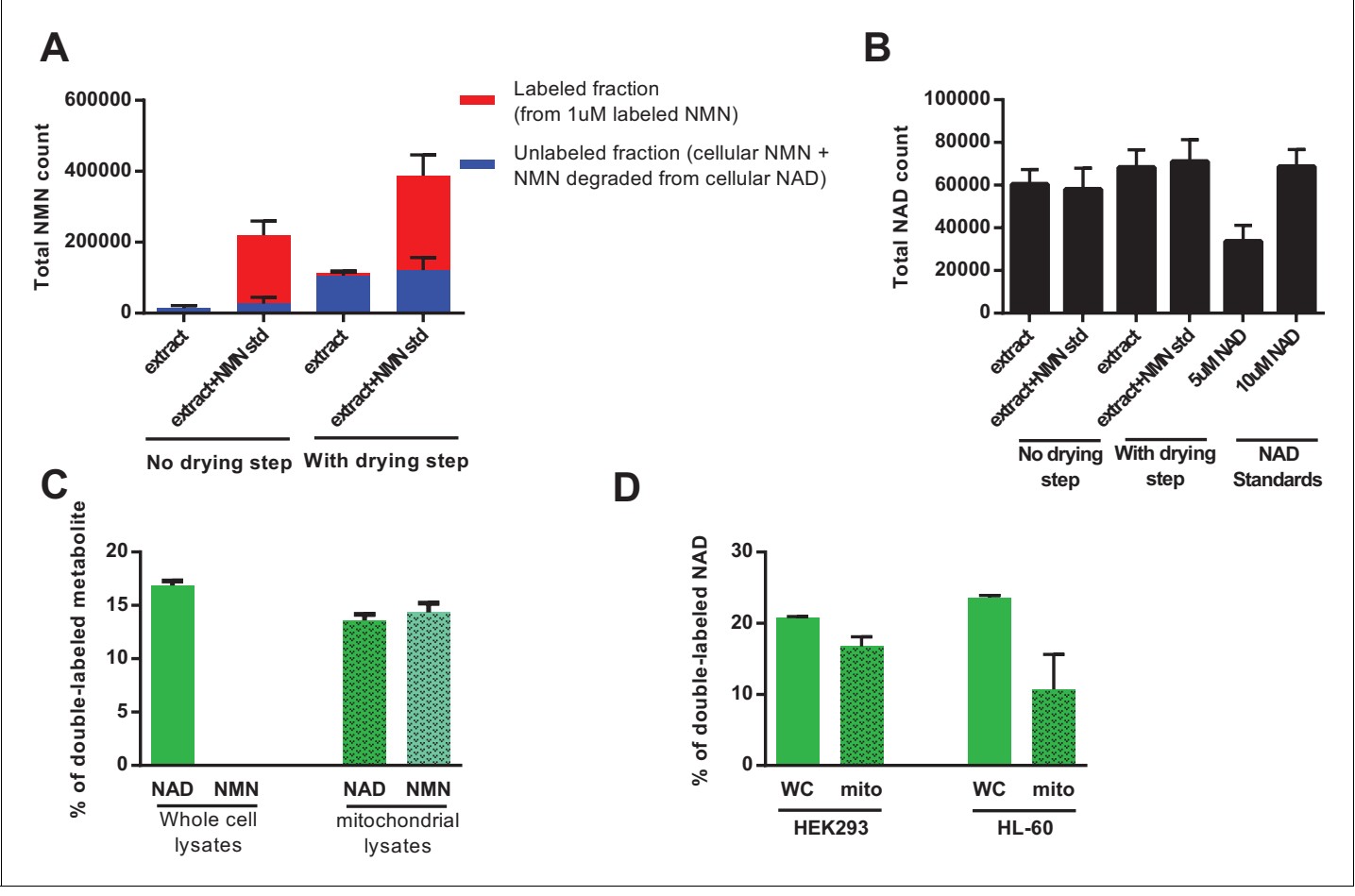

**Figure 7.** Direct injection of methanolic extracts reveals preferential labeling and mitochondrial uptake of NAD over NMN. (**A**) NMN concentration and labeling in differentiated C2C12 cells extracted with −80°C 80:20 methanol:water, analyzed either by hydrophilic interaction chromatography (no drying step), or dried under $N_2$, re-suspended in water and analyzed by reversed-phase ion-pairing chromatography (with drying step). Blue bars show unlabeled NMN resulting from intracellular NMN + NAD degradation after the drying/resuspension step; Red bars indicate labeled NMN from spiked-in standard (1 μM, dual labeled). (N = 2). (**B**) NAD total ion count measured in parallel from same samples in (**A**). (N = 2). (**C**) NAD and NMN labeling (mass + 2 fraction) in differentiated C2C12 cells treated with dual labeled NAR for 4 hr (whole cell vs. isolated mitochondria, N = 3). Data are compiled from three biological replicates and are displayed as means ± SEM. (**D**) NAD labeling (mass +2 fraction) in the human cell lines HEK293 and HL-60 following a 4 hr incubation with dual labeled NAR (whole cell vs. isolated mitochondria, N = 3). Data are compiled from three biological replicates and are displayed as means ± SEM.

DOI: https://doi.org/10.7554/eLife.33246.012

Our studies using isotopically labeled NR and NAR unequivocally demonstrate that the mitochondrial NAD pool can be established through direct import of NAD (or NADH). Using doubly labeled NR resulted in nearly equivalent labeling of NMN and NAD in whole cell lysates or mitochondria. The existence of doubly labeled NAD within the mitochondria in this experiment proves that mitochondrial NAD synthesis does not require nicotinamide import (as this would only carry a single label), but does not distinguish whether NMN or NAD was transported. To accomplish this, we differentially labeled NAD and NMN by providing doubly labeled NAR. NAR is converted to NAD in the cytosol via NAMN and NAAD, and therefore should not label the NMN pool. Our initial experiment was compromised by hydrolysis of a minority of the NAD to NMN during extraction, but when this technical hurdle was overcome, either by CRISPR targeting of NMNAT1 to enlarge the NMN pool or by directly injecting methanolic extracts into the LC-MS to avoid hydrolysis, the expected pattern of NAD labeling without NMN labeling was obtained. Under these conditions, the mitochondrial pool of NAD was also labeled, demonstrating that it originated from imported cytosolic NAD, rather than NMN. Although it remains technically possible that cytosolic NADH could be converted

to NMNH by Nudix hydrolase activity, then rapidly imported and converted to mitochondrial NADH without equilibrating with NMN, we were unable to detect labeling of NMNH in NAR-treated myotubes, and thus view this as a remote possibility.

Therefore, our results indicate that mammalian mitochondria contain an NAD or NADH transporter. While we are not the first to suggest that mitochondria can take up NAD(H), the identity of the putative transporter in mammalian mitochondria has never been elucidated and its existence continues to be debated. A number of proteins have been identified that allow NAD to cross membranes (*Bruzzone et al., 2001*; *Verderio et al., 2001*), but none of these have been shown to act in mitochondria (*Haferkamp et al., 2004*; *Palmieri, 2013*; *Haitina et al., 2006*). The known member of the mitochondrial solute carrier family that transports NAD, SLC25A17, has been localized exclusively to peroxisomes, where it functions to exchange NAD, FAD and free CoA for adenosine 3′,5′-diphosphate, FMN and AMP (*Palmieri, 2013*; *Agrimi et al., 2012*). In yeast and plants, nucleoside deoxyribosyltransferases transport NAD across the mitochondrial inner membrane from the cytosol by exchanging AMP and GMP or more slowly by uniport (*Palmieri, 2013*; *Todisco et al., 2006*). However, candidate mammalian NAD transporters identified based on sequence homology have proven to have alternative targets (e.g., the mitochondrial folate carrier) (*Yang et al., 2007*; *Haitina et al., 2006*; *Di Noia et al., 2014*). Recently, the plant mitochondrial NAD transporter, AtNDT2, was targeted and expressed in the mitochondrial membrane of human HEK293 cells, which resulted in the redistribution of cellular NAD into mitochondria (*VanLinden et al., 2015*). Surprisingly, this led to a slower proliferation, a significant reduction oxidative respiration and a dramatic loss of cellular ATP, which was attributed to a metabolic shift from oxidative phosphorylation to glycolysis (*VanLinden et al., 2015*). These results were interpreted to suggest that a mitochondrial NAD transporter is unlikely to exist in human cells. Nonetheless, our findings support the ability of mammalian mitochondria to import NAD and suggest that the toxicity of AtNDT2 may be more related to its specific kinetics or regulation than to a generalizable effect of NAD transport. Importantly, our findings do not exclude the possibility that NMN import and synthesis via NMNAT3 also contribute to the mitochondrial NAD pool. Indeed, Cambronne et al. recently employed a fluorescent biosensor to demonstrate that mitochondrial NAD levels are sensitive to depletion of either NMNAT3 (mitochondrial) or NMNAT2 (Golgi/cytosolic), implying that both NMN and NAD import contribute to the mitochondrial NAD pool (*Cambronne et al., 2016*). This observation suggests that a mitochondrial transporter for NMN may also await discovery. Alternatively, it is possible that NMNAT3 could function primarily to reverse NAD(H) hydrolysis or could work in combination with enzymes such as Nudt13 that generate NMN(H) (*Abdelraheim et al., 2017*).

In summary, we show that mammalian mitochondria are capable of directly importing NAD (or NADH). This finding strongly suggests the existence of an undiscovered transporter in mammalian mitochondria.

# Materials and methods

## Key resources table

| Reagent type (species) or resource | Designation | Source or reference | Identifiers | Additional information |
|---|---|---|---|---|
| Cell line (Mus musculus) | C2C12; myotubes; myoblasts; | ATCC | ATCC CRL-1772 | Mouse myoblast, mycoplasma negative |
| Cell line (Homo sapiens) | HL-60 | ATCC | ATCC CCL-240 | Human leukemia, authenticated by STR profiling, mycoplasma negative |
| Cell line (Homo sapiens) | 293; HEK293 | Gift from Morris Birnbaum's lab | | Human embryonic kidney, authenticated by STR profiling, mycoplasma negative |
| Strain, strain background (Mus musculus) | C57BL/6 mice | The Jackson Laboratory | 000664 | C57BL/6J |
| Recombinant DNA reagent | LentiCRISPR v2 (Lentiviral vector) | Addgene | 52961 | |

*Continued on next page*

*Continued*

| Reagent type (species) or resource | Designation | Source or reference | Identifiers | Additional information |
|---|---|---|---|---|
| Recombinant DNA reagent | psPAX2 (Lentiviral packaging plasmid) | Addgene | 12260 | |
| Recombinant DNA reagent | pMD2.G (Lentiviral envelope expressing plasmid) | Addgene | 12259 | |
| Recombinant DNA reagent | pLX304 (Gateway Lentiviral vector) | DNASU plasmid repository | NMNAT1 | Clone ID: HsCD00434593 |
| Chemical compound, drug | ATP | Sigma-Aldrich | A2383 | |
| Chemical compound, drug | Protease | Sigma-Aldrich | P5380 | Protease from Bacillus lichenformis |
| Chemical compound, drug | ADP | Sigma-Aldrich | A2754 | |
| Chemical compound, drug | Pyruvate | Sigma-Aldrich | P2255 | |
| Chemical compound, drug | Malate | Sigma-Aldrich | M1000 | |
| Chemical compound, drug | B-NMN | Sigma-Aldrich | N3501 100 MG | |
| Chemical compound, drug | PRPP | Sigma-Aldrich | P8296 | |
| Chemical compound, drug | FCCP | Sigma-Aldrich | C2920-10MG | |
| Chemical compound, drug | Oligomycin | Sigma-Aldrich | O4876-5MG | |
| Chemical compound, drug | Protease inhibitor cocktail (Sigma); PI | Sigma-Aldrich | P8340 | Protease inhibitor cocktail solution |
| Chemical compound, drug | Alcohol dehydrogenase | Sigma-Aldrich | A3263-150KU | |
| Chemical compound, drug | Diaphorase | Sigma-Aldrich | D5540-500UN | |
| Chemical compound, drug | Resazurin | Sigma-Aldrich | R7017 | |
| Chemical compound, drug | Flavin mononucleotide | Sigma-Aldrich | F6750 | |
| Chemical compound, drug | Nicotinamide; NAM | Sigma-Aldrich | 72345 | |
| Chemical compound, drug | Hexadinitrine | Sigma-Aldrich | 107689 | |
| Chemical compound, drug | Nicotinic acid; NA | Sigma-Aldrich | N4126 | |
| Chemical compound, drug | Perchloric Acid | Sigma-Aldrich | 244252 | |
| Chemical compound, drug | NAD; Nicotinamide adenine dinucleotide | Roche | 101127965001 | |
| Chemical compound, drug | Proteinase K; ProtK | Roche | 03115887001 | |
| Chemical compound, drug | Protease inhibitor cocktail (Roche) | Roche | 11697498001 | Complete protease inhibitor cocktail tablets |
| Chemical compound, drug | BSA | Roche | 03117057001 | Bovine serum albumin Fraction V, heat shock, fatty acid free |
| Chemical compound, drug | Puromycin | ThermoFisher Scientific | A111380-03 | 10 mg/mL stock |
| Chemical compound, drug | Blasticidin | ThermoFisher Scientific | R21001 | |
| Chemical compound, drug | Gallotannin | Enzo Life Sciences | ALX-270–418 G001 | |
| Chemical compound, drug | Insulin | Novo-Nordisk Novolin N | U-100 | 100 units/mL; recombinant DNA origin |
| Chemical compound, drug | Fugene 6 | Promega | E2691 | |
| Chemical compound, drug | NR; isotope-labeled nicotinamide riboside | PMID: 27508874 | | |
| Chemical compound, drug | NAR; isotope-labeled nicotinic acid riboside | this paper | | |
| Commercial assay or kit | Micro BCA Protein Assay Kit | Thermo Fisher Scientific | 23235 | |
| Commercial assay or kit | SuperSignal West femto kit | Thermo Fisher Scientific | 34095 | |

*Continued on next page*

*Continued*

| Reagent type (species) or resource | Designation | Source or reference | Identifiers | Additional information |
|---|---|---|---|---|
| Antibody | anti-NMNAT1 (rabbit polyclonal) | Gift from Lee Kraus, *Zhang et al. (2009)* PMID: 19478080 | | |
| Antibody | anti-VDAC (rabbit monoclonal) | Abcam | ab154856 | [EPR10852(B)] |
| Antibody | anti-B-actin HRP (mouse monoclonal) | Abcam | ab20272 | [mAbcam 8226] |
| Antibody | Secondary antibody | GE Healthcare Life Sciences | NA934 | Amersham ECL anti-rabbit IgG, HRP-linked whole Ab (from donkey) |
| Antibody | Secondary antibody | GE Healthcare Life Sciences | NA931 | Amersham ECL anti-mouse IgG, HRP-linked whole Ab (from sheep) |

## Mitochondrial isolation from skeletal muscle and liver

Male C57BL/6 mice were euthanized by cervical dislocation, and their gastrocnemius and quadriceps muscles were dissected and placed immediately in ice-cold muscle homogenization buffer (100 mM KCl, 50 mM Tris-HCl (pH 7.4), 5 mM $MgCl_2$, 1 mM EDTA (pH 8.0) and 1.8 mM ATP) at pH 7.2. The entire procedure was performed at 4°C. The fat and connective tissues were removed and the muscle tissue was chopped into small pieces. The chopped muscle was incubated for 2 min in protease medium (60U of protease from *Bacillus lichenformis* (Sigma) per mL of homogenization buffer), washed twice with homogenization buffer, and transferred to an ice-cold Teflon Potter Elvehjem homogenizer containing homogenization buffer. The muscle was homogenized using a motor-driven homogenizer for 10 min at 150 rpm. A small aliquot of the homogenate was then removed and stored at −80°C for further analysis. The volume of the remaining homogenate was doubled with homogenization buffer and centrifuged at 720 x g for 5 min at 4°C. The pellet was resuspended in homogenization buffer and centrifuged for an additional 5 min at 720 x g. The supernatants were combined and centrifuged at 10,000 x g for 20 min at 4°C. The supernatant was discarded and the pellet was resuspended in homogenization buffer and further centrifuged for 10 min at 10,000 x g. The final mitochondrial pellet was resuspended in resuspension buffer (225 mM sucrose, 44 mM $KH_2PO_4$, 12.5 mM Mg(OAc)$_2$, and 6 mM EDTA; pH 7.4) and maintained on ice. Mitochondrial protein content was quantified using the Micro BCA Protein Assay Kit (Thermo Scientific).

For isolation of liver mitochondria, the liver was placed into ice-cold mitochondrial isolation buffer [MIB; 210 mM Mannitol, 70 mM Sucrose, 10 mM HEPES, 1 mM EGTA, 0.25% BSA, pH 7.4 at 4°C] immediately after dissection, and chopped into small pieces. The liver was homogenized by 10 strokes in an ice-cold Teflon Potter Elvehjem homogenizer containing MIB. The homogenate was centrifuged at 750 g for 10 min at 4°C, and the resulting supernatant was centrifuged for a further 5 min at 750 g. The mitochondrial fraction was recovered by centrifugation of the supernatant for 20 min at 5000 g. The pellet was resuspended in homogenization buffer, and re-centrifuged at 5000 g for a further 10 min. The final mitochondrial pellet was resuspended in MIB lacking BSA for protein determination as described (Thermo Scientific).

## Mitochondrial treatments and extraction

For all experiments, purified mitochondria containing 100 µg of total protein were resuspended in ice-cold or pre-warmed MirO5 respiration buffer (Oroboros) containing the indicated compounds at a final concentration of 1 mg/mL. Pyruvate, Malate, ADP, β-NMN, PRPP, FCCP and Oligomycin were purchased from Sigma. NAD and NADH were from Roche. Gallotannin was from Enzo Life Sciences. For timed incubation experiments, the mitochondrial suspensions were maintained at 37°C in a shaking heat block with the tube caps opened. Proteinase K treatments (0.25 or 0.5 mg/mL, Roche) were performed for 30 min on ice with 1.1 mg/mL mitochondrial protein, then protease inhibitor cocktail (1:100, Sigma P8340) was added prior to warming. For NAD and NADH determination from the mitochondrial suspension, 50 µg of mitochondrial protein were transferred to tubes containing 10% (v/v) of either Perchloric Acid (Sigma) or KOH (Sigma) to achieve final concentrations of 0.6M or 0.1M, respectively. The mitochondrial lysates were vortexed vigorously, then centrifuged at max

speed for 10 min and the supernatant was collected for analysis and maintained on ice or stored at −70°C. Prior to storage or analysis, the KOH lysate were incubated at 55°C for 10 min to degrade any residual NAD, then cooled on ice for 5 min.

## NAD-NADH cycling assay

An enzyme-based cycling assay was used to determine NAD in experiments without tracers (*Figures 1–3*; *Figure 6a,b*). Immediately prior to analysis, mitochondrial lysates were diluted 1:10 in ice-cold phosphate buffer (pH 8). 5 µL of this dilution was then subjected to an enzymatic cycling assay in a 100 µL total volume as described previously (*Zhang et al., 2009*). Briefly, NAD standards or diluted mitochondrial extracts were added to a cycling mixture consisting of 2% ethanol, 100 µg/mL alcohol dehydrogenase, 10 µg/mL diaphorase, 20 µM resazurin, 10 µM flavin mononucleotide, 10 mM nicotinamide, 0.1% BSA in 100 mM phosphate buffer, pH 8.0. The cycling reaction was incubated at room temperature, and the appearance of resorufin (generated during each oxidation-reduction cycle) was measured by fluorescence excitation at 544 nm and emission at 590 nm.

## Cell culture

C2C12 myoblasts and HEK293 cells were cultured in Dulbecco's modified Eagle's medium (DMEM) supplemented with 4.5 g/L D-Glucose, 2 mM Glutamine, 10% FBS and antibiotics. C2C12s were purchased from ATCC and HEK293s were a gift from Morris Birnbaum's lab. Care was taken to maintain these cells within the log phase of growth and to avoid allowing them to become confluent. For differentiation into myotubes, C2C12s were grown to confluence, washed once with DPBS (Gibco) and the media was replaced with DMEM containing 2% Horse serum (Gibco) overnight. Following this, the media was replaced every day for 7 days with DMEM containing 2% Horse serum and 1 µM insulin (modified from [*Yaffe and Saxel, 1977*]). HL-60 cells were grown in cultured in Iscove's Modified Dulbecco's Medium (IMDM) supplemented with 20% FBS and antibiotics. HL-60 cells were purchased from ATCC. Care was taken to maintain these suspension cells at a concentration below $10^6$ cells per mL. Human cell lines (HEK293 and HL-60) were authenticated by STR profiling and all three cells lines tested negative for mycoplasma.

## Generation of CRISPR cell lines

The CRISPR/Cas9 system was used to target each of the three individual isoforms of NMNAT in C2C12 cells. For each isoform, two separate guide RNA sequences (gRNA) were targeted toward the 3' end of the coding region and were designed using the CRISPR design tool (http://crispr.mit.edu). A sequence from the ROSA 26 genes (R26) was used as a control. The gRNA sequences are listed in *Supplementary file 1*. The gRNAs were cloned into the LentiCRISPR v2 vector backbone (Addgene, #52961) between Esp3I sites downstream of the hU6 promoter. Lentivirus was produced by co-transfection of the lentiviral transfer vector with the pMD2-G envelope and psPAX2 packaging vectors into 293 cells using Fugene 6 transfection reagent (Promega). The media was changed 24 hr following transfection. The virus-containing supernatant was collected 48 hr post-transfection and filtered through a 0.22 µm syringe filter to eliminate cells. C2C12 myoblasts were infected with virus in media containing 8 µg/mL hexadinethrine (Sigma) in a dropwise manner with gentle swirling. 24 hr following infection, the virus was removed and the cells were selected in 1.5 µg/mL Puromycin (Gibco). Rescue of NMNAT1 targeted cells was accomplished by introduction of a lentiviral vector pLX304 (clone ID:HsCD00434593, DNASU, Arizona) expressing the human form of NMNAT1 (not targeted by the mouse-specific gRNA used to delete the gene). To generate active virus, the vector was co-transfected with the pMD2-G envelope and psPAX2 packaging vectors into 293 cells using Fugene six transfection reagent (Promega). Supernatant containing lentivirus was filtered through a 0.22 µm syringe filter, and used to infect cells in the presence of 8 µg/mL hexadinethrine (Sigma). After 24 hr, the virus was removed and the cells were selected in 3 ug/mL Blasticidin (Invitrogen).

## Western blot analysis

Whole cell or mitochondrial lysates from differentiated C2C12 cells were prepared with RIPA lysis buffer (50 mM Tris-HCl pH 7.4, 1% NP40, 0.25% sodium deoxycholate, 0.5 mM EDTA, 150 mM sodium chloride) supplemented with protease inhibitor cocktail (Roche). Forty µg of lysate, 20 µg of mitochondrial pellet, or 10 µg of supernatant were run on a 10% gel (Bio-Rad) and transferred to

PVDF membrane (Immobilon). The membrane was probed with rabbit polyclonal anti-NMNAT1 (1:500 dilution) as previously described (*Zhang et al., 2009*) or anti-VDAC (Abcam) followed by secondary antibody incubation. Immunoblots were developed using SuperSignal West femto kit (Thermo Fisher Scientific) on a Bio-Rad imaging system. Blots were then stripped and re-probed with HRP-conjugated β-actin antibody (Abcam).

## Synthesis of tracers

We designed double isotope-labeled nicotinamide riboside (NR) and nicotinic acid riboside (NAR) tracers, with a single $^{13}C$ and a single deuterium on the nicotinamide and ribose moieties, respectively (*Figure 4a*). Direct incorporation of the intact tracer into NAD yields double-labeled NAD, whereas breakdown and resynthesis by the salvage pathway of any cell yields single-labeled NAD (*Figure 4a*). The synthesis of the labelled NR was reported previously (*Frederick et al., 2016*). The synthesis of the $^{2}H$, $^{13}C$ NAR was accomplished as follows: $^{13}C$-Nicotinamide was hydrolysed under basic aqueous conditions to generate $^{13}C$-nicotinic acid, which following silylation was coupled to the 2D-tetraacetylated riboside under Vorbruggen conditions to yield the triacetylated $^{2}H$, $^{13}C$-NAR. Standard deprotection conditions employing $NH_{3g}$-MeOH at $-20°C$ for 4 days were employed to the generate $^{2}H$, $^{13}C$ NAR. $^{2}H$, $^{13}C$ NAR was isolated as a mixture of α/β anomers present in a 15:85 ratio, which could not be successfully separated. This α/β distribution proves reproducible, and is not observed for the non-labelled NAR ($^{1}H$ NMR, $^{13}C$ NMR, MS, HRMS). The $^{1}H$ NMR spectra of labeled and unlabeled NAR are provided in *Figure 5—figure supplement 1*. ESI-MS m/z 258.0926 (M + H); Exact mass calculated for ($^{12}C_{12}{}^{13}C_{1}{}^{1}H_{13}{}^{2}H_{1}N_{1}O_{6}$; M + H) 258.0917; found 258.0926.

## Isotopic labeling of cells

For the tracer studies, C2C12 myotubes, HEK293, or HL-60 cells were treated with double-isotope labeled 0.1 mM nicotinamide riboside (NR) or nicotinic acid riboside (NAR) in complete culture medium for 4 hr before extracting. The cells were then rapidly harvested using trypsin and media containing the label, were washed with ice-cold isolation buffer, and then 10% of the volume was removed and re-pelleted. To this pellet, 200 μL of 80:20 methanol:water (pre-chilled on dry ice) was added, vortexed vigorously and maintained on dry ice until processing as described below.

Mitochondria were isolated from the remaining 90% of the cells by a method modified from Trounce et al (*Trounce et al., 1996*). Briefly, the cells which were pelleted and resuspended in a mitochondrial isolation buffer (H-buffer) consisting of 210 mM mannitol, 70 mM sucrose, 1 mM EGTA, 5 mM HEPES, 0.5% BSA, pH 7.2. The cells were physically sheared in an ice-cold glass-glass dounce homogenizer then centrifuged at low-speed (720 x g for 10 min, 4°C). The supernatant containing the mitochondria was transferred to a separate tube, and pellet underwent a second round of homogenization and centrifugation. The supernatants were combined and further purified for the removal of cell debris through additional rounds of low speed spins. The resultant supernatant was subjected to two rounds of high-speed centrifugation (10,000 x g for 30 min total, 4°C). The resultant pellets of purified mitochondria were dissolved in cold resuspension buffer (225 mM sucrose, 44 mM $KH_2PO_4$, 12.5 mM Mg-acetate, and 6 mM EDTA; pH 7.4) and briefly spun (10,000 x g for 2 min, 4°C) in order to remove the mannitol from which interfered with the mass-spectrometry measurement.

Metabolism was quenched and metabolites were extracted by aspirating the wash buffer and immediately adding 500 μL 80% methanol (pre-chilled on dry ice). After 30 min of incubation on dry ice, the resulting mixture was centrifuged at 10,000 g for 5 min. The alcoholic supernatants were then subjected to LC-MS directly (*Figure 7*) or evaporated under nitrogen and resuspended in 200 μL water. Because mitochondrial fractions were prepared from nine times as much starting material as whole cell lysates, the ion counts obtained from mitochondria were divided by nine to facilitate a rough comparison. However, this method underestimates the mitochondrial contribution as there is some loss of material during isolation of the organelles. Quantification of NAM and NMN was performed by adding standard compounds to the solution.

In each experiment on isolated mitochondria, a single isolation was performed and the mitochondria subdivided into the indicated treatment groups. For experiments in unmodified cells, all experimental dishes were split from the same parental dish prior to the experiment. In the case of genetically modified cell lines, all of the cells to be compared were generated simultaneously from the same parent line, and were handled equivalently throughout the study.

## LC-MS instrumentation and method development

Nicotinamide, NMN, NR, and $NAD^+$, NAMN, NAR and $NAAD^+$ were analyzed within 24 hr by reversed-phase ion pairing chromatography coupled with positive-mode electrospray-ionization on a Q Exactive hybrid quadrupole-orbitrap mass spectrometer (Thermo); Liquid chromatography separation was achieved on a Poroshell 120 Bonus-RP column (2.1 × 150 mm, 2.7 μm particle size, Agilent). The total run time is 25 min, with a flow rate of 50 μL/min from 0 min to 12 min and 200 μL/min from 12 min to 25 min. Solvent A is 98: two water: acetonitrile with 10 mM ammonium acetate and 0.1% acetic acid; solvent B is acetonitrile. The gradient is 0–70% B in 12 min (*Lu et al., 2010*). All isotope labeling patterns were corrected for natural abundance using AccuCor with correction matrices calculated based on the chemical formula and the mass of the metabolite (*Su et al., 2017*).

## Statistics

Results are expressed as mean ± standard error of the mean of biological replicates. Biological replicates are individually treated samples (cells or isolated mitochondria) whereas technical replicates are repeat assays of the same biological replicate. No outliers were excluded from the presented data. No formal power analyses were used to design experiments. Sample sizes were chosen based on prior experience with similar assays. Comparisons between two groups were performed using Students *t* test to establish reported p-values. All statistical analyses were performed using Prism 6 (GraphPad Software, Inc).

## Acknowledgements

We thank all members of the Baur lab for constructive feedback and suggestions. NMNAT1 antibody was a kind gift from L Kraus. This work was supported by grants from the National Institutes of Health (R01-DK098656 and R01-AG043483 to JAB; K12-GM081259 to ADJ). We thank the University of Pennsylvania Diabetes Research Center (DRC) for the use of the Metabolomics Core (P30-DK19525).

## Additional information

### Funding

| Funder | Grant reference number | Author |
|---|---|---|
| National Institute of Diabetes and Digestive and Kidney Diseases | R01DK098656 | Joseph A Baur |
| National Institute on Aging | R01AG043483 | Joseph A Baur |
| National Institute of General Medical Sciences | K12DGM081259 | Antonio Davila Jr. |

The funders had no role in study design, data collection and interpretation, or the decision to submit the work for publication.

### Author contributions

Antonio Davila, Conceptualization, Formal analysis, Validation, Investigation, Visualization, Methodology, Writing—original draft, Writing—review and editing; Ling Liu, Conceptualization, Data curation, Validation, Investigation, Visualization, Methodology, Writing—review and editing; Karthikeyani Chellappa, Investigation, Writing—review and editing; Philip Redpath, Investigation, Methodology; Eiko Nakamaru-Ogiso, Validation, Investigation, Methodology, Writing—review and editing; Lauren M Paolella, Data curation, Validation, Methodology; Zhigang Zhang, Formal analysis, Investigation, Writing—review and editing; Marie E Migaud, Conceptualization, Formal analysis, Methodology, Writing—review and editing; Joshua D Rabinowitz, Conceptualization, Supervision, Funding acquisition, Methodology, Writing—review and editing; Joseph A Baur, Conceptualization, Resources, Data curation, Supervision, Methodology, Writing—original draft, Writing—review and editing

### Author ORCIDs
Antonio Davila (ID) http://orcid.org/0000-0003-1672-2713
Eiko Nakamaru-Ogiso (ID) http://orcid.org/0000-0003-0931-1940
Joseph A Baur (ID) http://orcid.org/0000-0001-8262-6549

### Ethics

Animal experimentation: This study was performed in strict accordance with the recommendations in the Guide for the Care and Use of Laboratory Animals of the National Institutes of Health. No live animal work was performed, and animals that were sacrificed for mitochondrial isolation were euthanized according to protocols approved by the institutional animal care and use committee (IACUC) of the University of Pennsylvania (protocol # 804892).

### Decision letter and Author response

Decision letter https://doi.org/10.7554/eLife.33246.019
Author response https://doi.org/10.7554/eLife.33246.020

## Additional files

### Supplementary files

• Supplementary file 1. Table of gRNA sequences cloned into LentiCRISPR v2 vector backbone. Two independent guides were used to target each NMNAT isoform, and were compared to a control targeting the ROSA26 locus.
DOI: https://doi.org/10.7554/eLife.33246.013

• Supplementary file 2. Primer sequences used to detect NMNAT transcripts. Each primer pair amplifies a region just downstream of the guide RNA that bears the corresponding name (see *Supplementary file 1*).
DOI: https://doi.org/10.7554/eLife.33246.014

• Transparent reporting form
DOI: https://doi.org/10.7554/eLife.33246.015

### Data availability

Source data for the figures has been submitted to Dryad (http://dx.doi.org/10.5061/dryad.qt58k)

The following dataset was generated:

| Author(s) | Year | Dataset title | Dataset URL | Database, license, and accessibility information |
|---|---|---|---|---|
| Davila Jr A, Liu L, Chellappa K, Redpath P, Nakamaru-Ogiso E, Zhang Z, Migaud ME, Rabinowitz JD, Baur JA | 2018 | Data from: Nicotinamide adenine dinucleotide is transported into mammalian mitochondria | http://dx.doi.org/10.5061/dryad.qt58k | Available at Dryad Digital Repository under a CC0 Public Domain Dedication |

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
