## [Decision Letter]

Thank you for submitting your article "Nicotinamide adenine dinucleotide is transported into mammalian mitochondria" for consideration by *eLife*. Your article has been reviewed by two peer reviewers, and the evaluation has been overseen by a Reviewing Editor and Ivan Dikic as the Senior Editor. The reviewers have opted to remain anonymous.

The reviewers have discussed the reviews with one another and the Reviewing Editor has drafted this decision to help you prepare a revised submission.

In this manuscript, Davila et al. provide evidence in support of the existence of a mammalian mitochondrial NAD (or NADH) transporter. This evidence is a combination of biochemical experiments, cell-based and purified mitochondria-based isotope-labeling experiments and includes the use of a couple of genetic loss-of-function models. The evidence is quite compelling in the systems that they employ. In that regard, this is a valuable contribution to the literature and is worth publishing.

There are few experiments that need to be conducted.

1) Why does NMN in the presence of ADP (state 3) increase NAD levels? State 2 is sufficient to develop the proton-motive force (pmf). If anything the addition of ADP slightly decreases pmf.

2) I am a bit unclear about NMNAT1 localization. The investigators argue it is in the isolated mitochondria preps. Do they envision that NMNAT1 is tethered to the outer mitochondrial membrane. Can they redo their assays with Proteinase K digestion that degrades exposed proteins on the outer mitochondrial membrane?

3) NMNAT1 CRISPR KO needs the appropriate control NMNAT1 cDNA rescue.

4) Can the authors provide data from human cells to support their claims? If not, please speculate upon conservation within the Discussion.

In summary, all reviewers were very favorable and kindly ask for the 4 points raised to be addressed.

---

## [Author Response]

There are few experiments that need to be conducted.1) Why does NMN in the presence of ADP (state 3) increase NAD levels? State 2 is sufficient to develop the proton-motive force (pmf). If anything the addition of ADP slightly decreases pmf.

We believe that the need for state 3 primarily reflects the requirement that ATP be available outside of the organelles. The reaction catalyzed by NMNATs is a condensation between NMN and ATP. Since our data suggest that the NMNAT reaction is occurring outside of the matrix, even any residual ATP trapped in the isolated organelles would not be available in state 2. However, the addition of ADP drastically increases the concentrations of all adenine nucleotides and allows the adenine nucleotide exchanger to transport ATP outside of the matrix. In support of this model, we find that adding ATP directly is sufficient to support NAD synthesis without the organelles ever actually entering state 3. We have added these data to Figure 1 (new panel F).

2) I am a bit unclear about NMNAT1 localization. The investigators argue it is in the isolated mitochondria preps. Do they envision that NMNAT1 is tethered to the outer mitochondrial membrane. Can they redo their assays with Proteinase K digestion that degrades exposed proteins on the outer mitochondrial membrane?

Given that several groups have localized NMNAT1 only to the nucleus and not to the mitochondria by tagging/immunofluorescence (Zhang et al., 2012; Berger et al., 2005), we view it as most likely that the presence of the enzyme in our preps reflects nuclear contamination, rather than tethering to the outer membrane. As suggested, we repeated the assays with Proteinase K digestion. Proteinase K reduced the ability of mito preps to convert NMN to NAD without impairing respiratory capacity, consistent with the model that NMNAT1 is located outside of the matrix. These data have been added as Figure 6—figure supplement 2.

3) NMNAT1 CRISPR KO needs the appropriate control NMNAT1 cDNA rescue.

The lentiviral CRISPR/Cas9 strategy that we used to knock NMNAT1 in C2C12 murine myocytes can result in stable expression of the Cas9 and guide RNA. Thus, we sought to rescue the knockout cells by overexpressing the human NMNAT1 cDNA which should be resistant to the mouse-specific NMNAT1 gRNA. Human NMNAT1 cDNA significantly attenuated the increase in NMN concentration in cells lacking murine NMNAT1 and enhanced the ability of mitochondrial preps to synthesize NAD from NMN. These data have been added as Figure 6—figure supplement 3.

4) Can the authors provide data from human cells to support their claims? If not, please speculate upon conservation within the Discussion.

We repeated the isotopic labeling experiments using human HEK293 and HL-60 cell lines. The cells were labeled with double-labeled nicotinic acid riboside (NAR) for four hours and metabolites were extracted from whole cells or mitochondrial lysates. In both cases, this strategy resulted in the appearance of doubly labeled NAD within the mitochondria, similar to the results in murine C2C12s. Unfortunately, both of the human cell lines had extremely low NMN levels that did not allow us to provide confident measurements of labeled fraction or concentration. However, given the very low abundance of NMN and knowing that NAR is not expected to generate NMN, these results again support the conclusion that NAD generated in the cytosol is taken up into mitochondria. These results have been added to Figure 7 (new panel D).